# Sexual Dimorphism of Ethanol-Induced Mitochondrial Dynamics in Purkinje Cells

**DOI:** 10.3390/ijms252413714

**Published:** 2024-12-22

**Authors:** Rehana Khatoon, Jordan Fick, Abosede Elesinnla, Jaylyn Waddell, Tibor Kristian

**Affiliations:** 1Department of Anesthesiology and the Center for Shock, Trauma and Anesthesiology Research (S.T.A.R.), University of Maryland School of Medicine, 685 Baltimore St., Baltimore, MD 21201, USA; rkhatoon@som.umaryland.edu (R.K.); aelesinnla@som.umaryland.edu (A.E.); 2Veterans Affairs Maryland Health Center System, 10 North Greene Street, Baltimore, MD 21201, USA; jfick@som.umaryland.edu; 3Department of Pediatrics, University of Maryland School of Medicine, 655 W. Baltimore St., Baltimore, MD 21201, USA; jwaddell@som.umaryland.edu

**Keywords:** mitochondria, cerebellum, Purkinje cells, alcohol, free radicals

## Abstract

The cerebellum, a key target of ethanol’s toxic effects, is associated with ataxia following alcohol consumption. However, the impact of ethanol on Purkinje cell (PC) mitochondria remains unclear. To investigate how ethanol administration affects mitochondrial dynamics in cerebellar Purkinje cells, we employed a transgenic mouse model expressing mitochondria-targeted yellow fluorescent protein in Purkinje cells (PC-mito-eYFP). Both male and female PC-mito-eYFP mice received an intraperitoneal injection of ethanol or vehicle. One hour after ethanol administration, the animals were perfusion fixed or their cerebellum tissue or isolated mitochondria were collected. Cerebellum sections were analyzed using confocal microscopy to assess changes in mitochondrial length distribution. In vivo superoxide levels were measured using dihydroethidium (DHE), and mitochondrial NAD levels were determined by high-performance liquid chromatography (HPLC). Our findings revealed a sex-dependent response to ethanol administration in mitochondrial size distribution. While male Purkinje cell mitochondria exhibited no significant changes in size, female mitochondria became more fragmented after one hour of ethanol administration. This coincided with elevated phosphorylation of the fission protein Drp1 and increased superoxide production, as measured by DHE fluorescence intensity. Similarly, mitochondrial NAD levels were significantly reduced in female mice, but no changes were observed in males. Our results demonstrate that ethanol induced mitochondrial fragmentation through increased free radical levels, due to reduced NAD and increased p-Drp1, in PC cells of the female cerebellum.

## 1. Introduction

Ethanol, even at low concentrations, elicits cerebellum-mediated motor alterations [1]. Consumption of alcohol results in ataxia due at least in part to its effects on the cerebellum. Although the cerebellum contains multiple types of neurons, Purkinje neurons give rise to the single output of the cerebellar cortex [2]. Therefore, dysfunction or altered functional outcome of these neurons inevitably provokes impairment of motor behavior and motor learning.

The physiological function of neurons and their communication is supported by the cellular bioenergetic and neurotransmitter metabolism that requires corresponding distribution and movement of functional mitochondria throughout the cell soma and cellular processes [3]. This is to ensure an efficient supply of ATP wherever it is needed. Consequently, the two major functions of mitochondria, the generation of ATP and reorganization of their sub-cellular distribution via movement and fission/fusion dynamics, are essential mechanisms for adaptive cellular response under stress conditions, including ethanol-induced stress [4,5]. Mitochondrial dynamics are orchestrated and controlled by proteins which regulate the fission and fusion process [6,7,8,9]. The major protein controlling fission is dynamin-relate protein 1 (Drp1). When phosphorylated at Ser 616, p-Drp1 initiates the fission process [10].

This study characterized changes in mitochondrial dynamics after acute ethanol administration. We focused on the cerebellum, particularly Purkinje cell mitochondria, because of its susceptibility to ethanol [11,12,13].

## 2. Results

### 2.1. Distribution of Mitochondria in Purkinje Cells

To achieve the expression of enhanced yellow florescent protein (eYFP) in neuronal mitochondria (mito-eYFP), we crossed two transgenic mouse models. One expresses tetracycline trans-activator (tTA) driven by a neuron specific promoter, calcium/calmodulin-dependent kinase IIα (Camk2α), and the other carries a gene for tetracycline response element (TRE) promoter regulated mitochondria-targeted eYFP (TRE-mito-eYFP) [14]. Transgenic offspring that carry both the TRE-mito-eYFP and Camk2α-tTA gene express the mitochondria-targeted eYFP in neurons [14,15]. As we reported, the mito-eYFP expression does not affect mitochondrial respiratory function and does not show any adverse or toxic effects [14]. The mito-eYFP expression is observed in cortical and hippocampal neurons, and furthermore, there is a strong expression observed in striatum [14,15]. Since Purkinje cells also express Camk2α [16,17], we can detect the mito-eYFP within the Purkinje neurons’ mitochondria (Figure 1). Immunostaining of the cerebellar tissue sections with calbindin antibody revealed that the mito-eYFP expression was heterogeneous. In many Purkinje cells, the eYFP levels were very low or it was not detected by confocal microscopy (Figure 1C). We do not know the reason for such diversity in the eYFP expression levels. We observe similar heterogeneity of eYFP expression in astrocytes of another transgenic mouse model in which mito-eYFP expression is regulated by the glial fibrillary acidic protein (GFAP) promoter [15]. Cerebellar sections were used to collect Z-stack images from Purkinje neurons with strong mito-eYFP expression. As Figure 1 and Figure 2 show, mitochondria in Purkinje cells display unique spatial distribution and density. The perinuclear region contains a high density of mitochondrial organelles. Within the axon, the mitochondria are short tubular structures organized along the axon trajectory and distributed evenly throughout the soma. The dendritic arbor is filled with mitochondria that follow the shape of the complex structure of the processes. Thus, mitochondria form a highly interconnected tubular network throughout the structure of the Purkinje neuron. The densely packed mitochondria in the perinuclear region limits the ability of the confocal microscopy to distinguish between individual organelles. The mitochondria in the dendritic branches are elongated and organized longitudinally in the processes.

### 2.2. Acute Effect of Ethanol Intake on Purkinje Cell Mitochondrial Dynamics

One hour following ethanol administration, the serum ethanol concentration in males was 103.1 ± 0.8 mM, and in females, the ethanol levels were significantly higher at 145.6 ± 2.3 mM. Figure 3 shows a representative image reconstructed from z-stack serial sections of Purkinje neuron mitochondria and quantification of their length distribution for both male and female cerebella. Although the ethanol-induced fragmentation is not readily observable without quantitative analysis, quantification did reveal a significant fragmentation of the mitochondrial population in female Purkinje cells (Figure 3C,D). Male Purkinje mitochondria in the 15–40 μm range trended toward a reduced number (*p* = 0.099) with a corresponding increase in the relative count within the 5–15 μm range (*p* = 0.196), but these differences between the vehicle (control) and ethanol-treated group were not significant (Figure 3A,B).

In female PC, the relative number of mitochondria within the 5–15 μm and 15–40 μm ranges decreased (*p* = 0.026 and *p* = 0.045) while the number of shorter mitochondria (0.2–1 μm and 1–5 μm) increased (*p* = 0.0012, and *p* = 0.047) (Figure 3D).

The major regulator of mitochondrial fission is the phosphorylated form of dynamin-related protein 1 (Drp1) at serin 616 (p-Drp1(ser616)). Figure 4 shows the effect of ethanol administration on the levels of p-Drp1(ser616) in male and female cerebella following different time periods of post-ethanol intake. Interestingly, p-Drp1(Ser616) levels were significantly increased 1 h after ethanol intake only in females.

### 2.3. Ethanol Intake Leads to Increased Production of ROS in Female Purkinje Cells

One factor that can trigger mitochondrial fragmentation is increased generation of reactive oxygen species (ROS). We therefore used the in vivo superoxide indicator dihydroethidium (DHE) to detect changes in cerebellum ROS production after ethanol administration.

Once DHE reacts with superoxide anions, a red fluorescent ethidium is formed [18,19]. Figure 5 show punctate staining, indicating that ethidium florescent was detected mainly in the perinuclear regions of Purkinje cells, where the mitochondrial density is high. Interestingly, ethanol administration significantly increased the signal intensity only in female PC (Figure 5 right panel). There was no significant effect of ethanol on the ethidium signal in male mice (Figure 5 left panel).

### 2.4. Acute Ethanol Effect on Mitochondrial NAD Levels

Free radical levels are determined not only by the rate of ROS generation but also by the rate of superoxide detoxification. Since NAD can modulate activity of superoxide dismutase by altering its acetylation via NAD-dependent deacetylase sirtuin 3 (Sirt3) [20,21], we determined the effect of ethanol consumption on mitochondrial NAD levels. Figure 6 shows changes in isolated mitochondria NAD pools from male and female cerebella following ethanol administration. Interestingly, NAD levels were significantly reduced only in female cerebellar mitochondria from 3.5 nmol/mg to 2.6 nmol/mg mitochondrial protein.

## 3. Discussion

In this study, we used transgenic mice that express mitochondria-targeted eYFP (mito-eYFP) to visualize mitochondria within Purkinje cells and examined the effect of ethanol intake on mitochondria dynamics. Previous studies that described Purkinje cell mitochondria also used transgenic animals expressing mitochondria-targeted fluorescent markers (mito-GFP, mito-Dendra2) [22,23]. In an alternative approach, animals received an injection of adenovirus encoding mito-eGFP into the molecular layer of the cerebellum to visualize and quantify the morphologic parameters of mitochondria [22,24]. In another study, PC mitochondria in wild-type mice and in mice with autosomal recessive mutation were characterized using electron microscopy [25]. Our data show a uniquely high density of mitochondria distributed throughout the intracellular volume, including the abundantly branched processes (see also [24,26,27]). Here we report that mitochondria in PC form a complex interconnected network of organelles with lengths up to 40 μm. The long tubular mitochondria were identified particularly in the dendritic tree. This is in agreement with data reported by Li et al. (2023) using the mito-GFP-expressing mouse model. In their work, the authors stated that the biggest mitochondria were in the dendritic tree with a surface area of 45.3 ± 53.21 µm^2^ [22]. Considering the cylindric shape of mitochondria and an approximately 0.8 µm diameter, the longest organelles were up to 40 µm. In previous studies, we used our transgenic mouse model to characterize mitochondria in hippocampal or cortical neurons [14,15,21]. In these cells, the longest mitochondria were about 15 μm [15,21]. Thus, Purkinje cell mitochondria display uniquely long and complex shapes.

Mitochondria play a crucial role during the maturation of Purkinje cells since they deliver the required energy for dendritic arborization [25,26]. An abundant number of mitochondria is required due to extensive consumption of ATP by these cells for active ion transport to maintain their spontaneous firing activity [28,29,30]. The high demand of Purkinje cells for ATP generation is also due to their role as the sole output neurons in the cerebellar circuit, as inputs from parallel and climbing fibers are integrated along their highly complex and intricate dendrites [31].

Following alcohol intake, most of the ethanol is oxidized by the liver. However, ethanol readily passes through biological membranes and readily distributes into all organs, including the brain [32]. It is widely established that ethanol metabolism generates reactive oxygen species (ROS) [33,34,35,36]. Oxidation of ethanol results in substantial reduction of NAD to NADH, thus increasing the NADH/NAD ratio. This generates a strong reducing pressure that increases ROS generation by mitochondria [37]. Ethanol-induced oxidative stress in brain tissue has been studied mainly under conditions of chronic ethanol intake or following ethanol withdrawal. These conditions lead to significant lipid peroxidation and depletion of mitochondrial DNA that has been associated with inhibition of mitochondrial respiration [33,36,38,39]. These pathologic outcomes can be reversed by antioxidants or inhibition of alcohol dehydrogenase [39]. We detected a significant increase in superoxide generation in female Purkinje cells, but did not observe a significant change in male cerebellum following ethanol administration. To our knowledge, previously published studies with acute ethanol effects on ROS production in brain tissue did not include comparison between male and female samples. The increase in the levels of ROS can result from either a higher rate of ROS production or due to the inhibition of enzymes that detoxify ROS [40]. The strong reducing pressure and high NADH levels due to ethanol catabolism suggest that mitochondria will generate more superoxide. Superoxide in mitochondria is detoxified by superoxide dismutase 2 (SOD2) [40,41,42]. The activity of SOD2 is modulated by acetylation, which is controlled by mitochondrial NAD-dependent deacetylase SIRT3 [40,43]. Increased acetylation inhibits SOD2 activity, which leads to increased superoxide levels [20,44]. NAD levels in female cerebellum were significantly reduced due to alcohol administration [45]. Therefore, it is likely that the increase in ROS levels following ethanol administration is higher in females than in males.

Oxidative stress activates the phosphorylation of Drp1 and its translocation to the outer mitochondria membrane leading to mitochondrial fission [40,44]. Thus, coincident increased superoxide generation and higher p-DRP1 levels cause increased fragmentation of mitochondria in female PC.

Our data suggest slower ethanol metabolism in females compared to males. This results in prolonged higher ethanol levels in female animals, and it may lead to differences in downstream ethanol effects on mitochondria. Further work is necessary to compare blood alcohol levels between the sexes to determine whether these differences in rate of metabolism fully explain our observed sex differences.

In conclusion, this study shows that ethanol metabolism in cerebellum has differential effects on mitochondrial dynamics with a stronger impact in females that is probably due to reduction in mitochondrial NAD pools, leading to an increase in superoxide levels and, consequently, mitochondrial fragmentation.

## 4. Materials and Methods

### 4.1. Animals

All animal experiments were performed in accordance with the Guide for the Care and use of Laboratory Animals of the National Institute of Health and were approved by the IACUC of the University of Maryland Baltimore. Adult, 3-month-old C57Bl6 male and female wild-type (WT) mice (Jackson Laboratory, Main, USA) and transgenic mice that express yellow fluorescent protein targeted to mitochondria (mito-eYFP) in neurons [14,15] were used for experiments. In this transgenic model, mito-eYFP expression is driven by the calcium/calmodulin-dependent kinase 2α (Camk2α) promotor. The animals were maintained in a 12 h light/dark cycle and were housed in groups of 2 to 5 mice per cage in a temperature-controlled room at 23 °C. The mice were free of all viral, bacterial, and parasitic pathogens. All mice were randomly assigned to experimental groups. Male and female ethanol-treated mice were injected with 20% ethanol in PBS at a dose of 2 g/kg. Control mice received an equal volume of PBS (vehicle) intraperitoneally (i.p.).

### 4.2. Measurements of Serum Ethanol Concentration

Serum ethanol concertation was determined by using an alcohol assay kit from Abnova (KA4784, Taipei City, Taiwan). One hour after the ethanol administration, blood samples were collected in a tube with no anticoagulant after decapitation. The blood was allowed to clot at room temperature for 30 min. This was followed by centrifugation of the sample at 2500× *g* for 20 min. Then, the yellow serum supernatant was collected without disturbing the white buffy layer. The serum samples were then stored at −80 °C before the determination of ethanol concentration.

### 4.3. Isolation of Mitochondria from Cerebellum

Non-synaptic mitochondria were isolated from the mouse cerebellum using a percoll gradient centrifugation, as described in [21,46]. Briefly, 1 h after ethanol administration, mouse brains were removed from the skull following decapitation, and the cerebellum was dissected on ice. Then, the tissue was homogenized in 15% percoll prepared by dilution with isolation medium (225 mM sucrose, 75 mM mannitol, 1 mM EGTA, 5 mM Hepes, pH 7.0, MilliporeSigma, St. Louis, MO, USA) containing deacetylase inhibitors (10 mM TSA, 10 mM nicotinamide, 10 mM sodium butyrate, MilliporeSigma, USA). The homogenate was layered on the 24–40% percoll gradient and centrifuged at 30,700× *g* for 15 min at 4 °C. Mitochondrial layer at the 24% and 40% percoll interface was collected, resuspended in isolation media, and centrifuged at 16,700× *g* for 10 min at 4 °C. Supernatant was discarded, and the pellet was again resuspended in the isolation media and centrifuged at 6900× *g* for 10 min at 4 °C. The final pellet, representing purified non-synaptic mitochondria from the cerebellum, was used for NAD content determination.

#### Measurement of Mitochondrial NAD Levels by HPLC

NAD in isolated mitochondrial samples was determined after using perchloric acid (PCA) extraction procedure [44]. The mitochondrial sample was mixed with 7% PCA and incubated on ice for 15 min. This was followed by centrifugation at 10,000× *g* for 10 min at 4 °C. The pellet was used for total protein determination by Lowry assay, and the supernatant was neutralized with 1M Trizma and 9 M KOH. Following neutralization, the extract was centrifuged again at 10,000× *g* at 4 °C, and the supernatant was analyzed by Agilent 1260 Infinity 11 LC high-performance liquid chromatography (HPLC) (Agilent, Santa Clara, CA, USA) with a C18 (250 mm × 4.6 mm, 5 micron) column [44]. To the extracted metabolites from cerebellum tissue samples, a concentration gradient similar to [47] (see also [48]) was created with 50 mM sodium phosphate pH 6 and 50% of 50 mM sodium phosphate in HPLC-grade methanol, pH 7. The gradient consisted of the following steps (in % methanol): 0 min, 0%; 2.5 min, 0.5%; 5 min, 3%; 7 min, 5%; 8 min, 12%; 10 min, 15%; 12 min, 20%; 20 min, 30%. The authenticity of NAD in the samples was confirmed by co-elution with added NAD standard. All samples were diluted 1:1 in their respective 50 mM sodium phosphate buffer before being loaded and run on HPLC.

### 4.4. Immunohistochemistry

Mice were perfusion fixed under deep anesthesia one hour after ethanol or vehicle administration. Once anesthetized, mice were intubated, ventilated, and then transcardially perfused for 1 min with oxygenated cold (4 °C) PBS. This was followed by perfusion with warm (37 °C) same-day-prepared 4% paraformaldehyde [15,21]. Following perfusion fixation, the brains were removed from the skull and post-fixed in paraformaldehyde at 4 °C for 24 h. Then, the brains were transferred into 30% sucrose for 2 days. Sagittal sections of 40 µm from cerebellum were cut on freezing microtome and collected. The sections were stored in cryoprotectant solution at −20 °C before processing for immunostaining. The tissue sections were washed with KPBS and then incubated with Calbindin D-28k (Swant AG, Burgdorf, Switzerland 1:2000) overnight at 4 °C in 0.3% KPBS-T (Triton X-100, MilliporeSigma, USA). This was followed by washing the sections in KPBS and incubation in goat anti-rabbit (Alexa Fluor 594, #A-11012, Thermo Fisher Scientific, Waltham, MA, USA) secondary antibody (1:600) for 1 h at room temperature. The sections were examined with Keyence BZ-X710 (Keyence Corporation of America, Itasca, IL, USA) fluorescence microscope.

### 4.5. Superoxide Detection In Vivo

For in vivo reactive oxygen species (ROS) studies, we administered dihydroethidium (DHE), which is oxidized to fluorescent ethidium by superoxide [49]. DHE was prepared as a 1 mg/mL solution in DMSO and administered 2 mg/kg by i.p. injection. DHE was injected 30 min before vehicle or ethanol administration. One-hour post-ethanol or vehicle injection, the animals were perfusion fixed with 4% paraformaldehyde, and the brain was removed and processed for histology. Brain sections were mounted on a slide, and images from the cerebellum were collected. Four images were taken from each section. The total fluorescence intensity from the Purkinje cell layer was determined by Volocity 6.3.1 software (Quorum Technologies Inc., Puslinch, ON, Canada) and normalized to area.

### 4.6. Western Blots

The cerebellum was dissected on ice and homogenized in lysis buffer (NaCl 150 mM, Tris 10 mM, 1% Triton X-100, 0.5% nonidet p-40) with protease inhibitor cocktail (MilliporeSigma, USA). Twenty-five nanograms of protein were gel-separated and transferred to immobilon PVDF-FL membrane. Membranes were incubated in Odyssey blocking buffer (Licor Biosciences, Lincoln, NE, USA) for an hour at room temperature. This was followed by incubation of membranes with primary antibody overnight at 4 °C (anti-Drp1:1000 cell signaling #14647S; anti-pDrp1(Ser616) 1:1000 cell signaling #4494). After washing the membranes in PBS with 0.1% tween-20, they were incubated with the appropriate infrared fluorophore conjugated secondary antibody (Licor Biosciences, Lincoln, NE, USA) for 30 min at room temperature. The membranes were scanned by Sapphire FL biomolecular imager from Azure biosystems, and the bands were quantified using AzurSpotPro 1.4 software.

### 4.7. Mitochondrial Dynamics Quantification

Mitochondrial length in vehicle- and ethanol-treated mito-eYFP mice was determined 1 h after the treatment. Z-stack images of mitochondria from the PC layer (4 images per section) were collected using a laser scanning confocal microscope and quantified by Volocity software, as described in [15,44]. The software identifies individual mitochondria, reconstructs their 3D shape, and measures morphometric parameters of individual objects, including skeletal length. After the morphometric data were obtained, mitochondria were sorted based on length and divided into several groups with defined length intervals [15,44]. Final calculations were conducted in Excel, and graphs were constructed using Prism10 (GraphPad, Boston, MA, USA).

### 4.8. Laser Scanning Confocal Microscopy

To collect z-stack series of images from cerebellar sections, a Ziess LSM 510 laser scanning confocal microscope using a Plan-Apochromat 63×/1.4 oil lens was used (Ziess, White Plains, NY, USA). Single planes of 1024 × 1024 pixels were recorded at 1.0–1.5 Airy unit pinholes every 0.2 μm z-spacing throughout the whole tissue section. The z-stack images were obtained from the dendritic tree of Purkinje cells.

### 4.9. Statistics

Data are expressed as means ± standard error of mean (SEM). Statistical analysis was performed using Prism (GraphPad) version 10. Statistical significance was assessed by Student *t*-test when two groups were compared or one-way ANOVA for multiple groups. The *p* values < 0.05 were considered statistically significant.

## Figures and Tables

**Figure 1 ijms-25-13714-f001:**
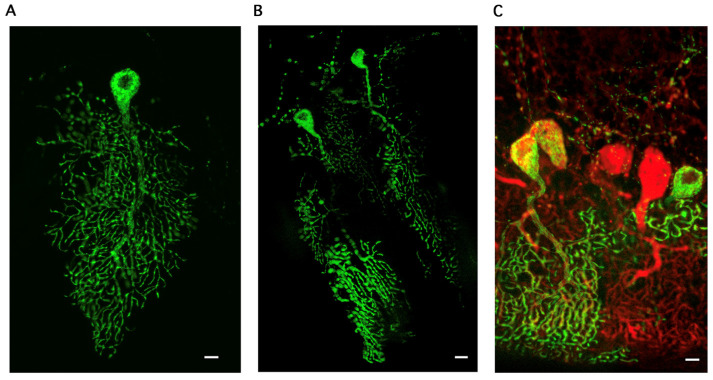
Mitochondrial network in Purkinje neurons. Transgenic mouse model that expresses mitochondria-targeted eYFP, driven by Camk2α promotor, was used to visualize mitochondria in Purkinje neurons. Panels (**A**,**B**) represent epifluorescent images of mitochondria within the Purkinje cells’ (PCs) soma and dendritic processes. Panel (**C**) shows staining of PCs with calbindin antibody (red). Scale bar represents 10 µm.

**Figure 2 ijms-25-13714-f002:**
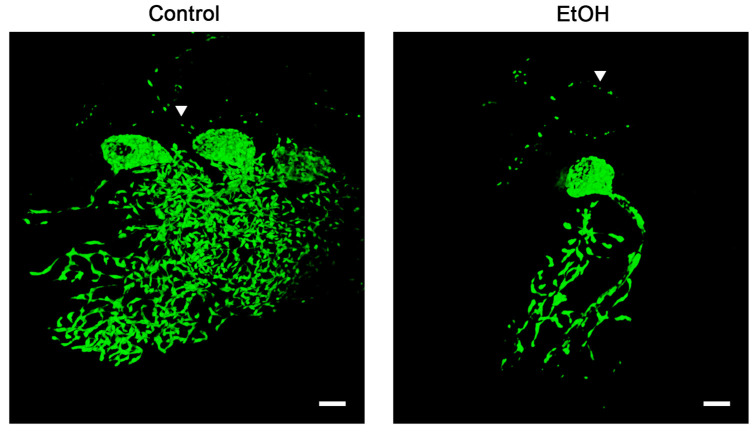
Purkinje cell mitochondria. Image generated from z-stack confocal images. Control, vehicle-treated male mouse (**left** panel), and following ethanol administration (**right** panel). Triangle indicates axonal mitochondria distributed along the axonal axes. Scale bar represents 10 µm.

**Figure 3 ijms-25-13714-f003:**
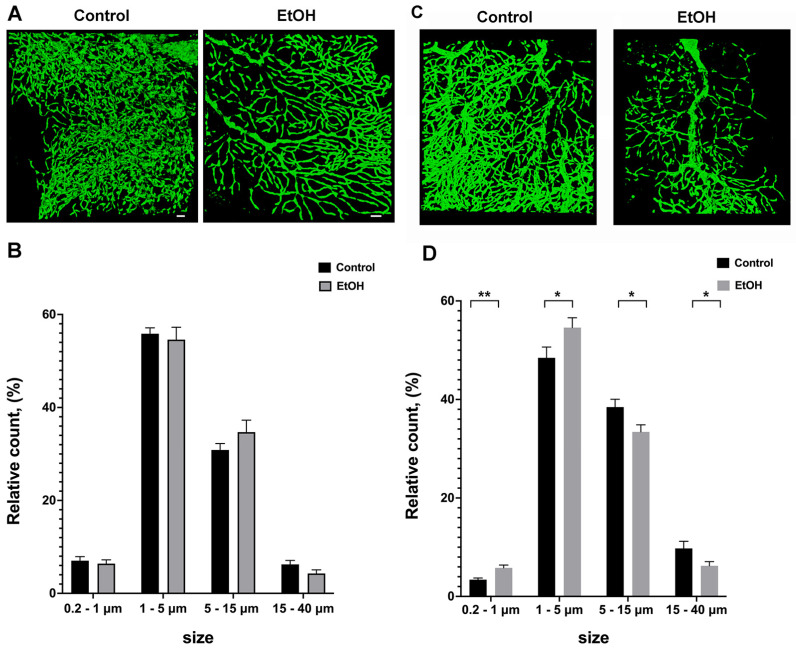
Ethanol metabolism leads to increased fragmentation of female PC mitochondria. (**A**) Images of mitochondrial network within dendritic tree of control (left panel (**A**)) and ethanol-treated male mouse (right panel (**A**)). (**B**) Quantification of mitochondrial size distribution in control PC and 1 h after ethanol administration using Volocity 6.3.1 software. There were no significant changes in relative number of mitochondria in the populations of 0.2–1 μm, 1–5 μm, 5–15 μm, and 15–40 μm length groups after ethanol administration. One-way ANOVA for multiple groups (n = 16 images/group). (**C**) Representative images of PC mitochondria from control and ethanol-treated female mouse cerebella. (**D**) Quantification of the changes in relative length distribution 1 h following alcohol administration. The relative distribution of individual mitochondria populations shows that ethanol administration increased the number of short mitochondria (0.2–1 μm and 1–5 μm) when compared to control. Conversely, in the longer mitochondria population (5–15 μm and 15–40 μm), the relative number decreased. ** *p* < 0.01; * *p* < 0.05 One-way ANOVA for multiple groups (n = 16 images/group). Scale bar represents 10 µm.

**Figure 4 ijms-25-13714-f004:**
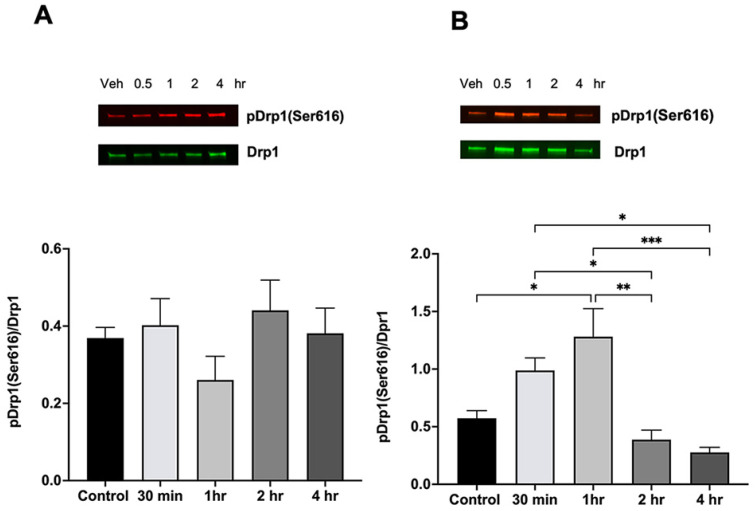
Ethanol administration increases the p-Drp1(Ser6161) levels in female cerebellum. (**A**) p-Drp1(Ser616)-to-Drp1 ratio in male cerebellum following ethanol treatment. There was no significant effect on Drp1 phosphorylation levels induced by ethanol metabolism. (**B**) In female cerebellum, the p-Drp1 levels were increased at 30 min and 1 h after the ethanol administration. * *p* < 0.05, ** *p* < 0.01, *** *p* < 0.001, n = 4.

**Figure 5 ijms-25-13714-f005:**
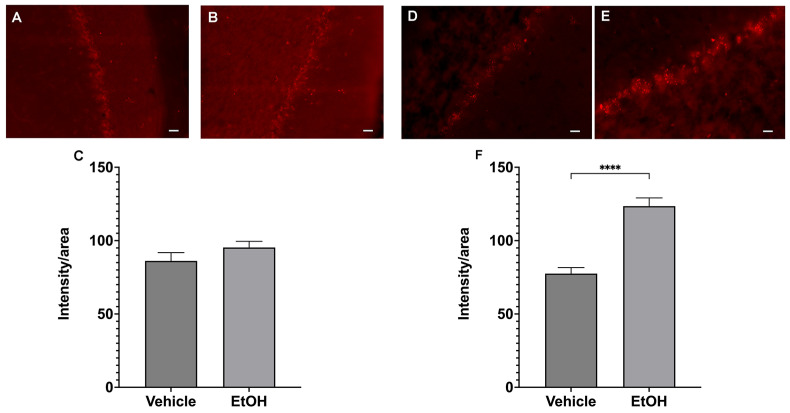
Ethanol administration increases superoxide levels in female cerebellum. Left panel shows ethidium fluorescence in control (**A**) and after ethanol (**B**) in male cerebellum. Right panel shows images of ethidium fluorescent taken from female cerebellum sections ((**D**) control vehicle, (**E**) ethanol). (**C**,**F**) Quantification of fluorescent intensity normalized to area. There is a significant increase in florescent intensity in female cerebellum following ethanol administration. **** *p* < 0.001 when compared to vehicle, *t*-test, n = 12.

**Figure 6 ijms-25-13714-f006:**
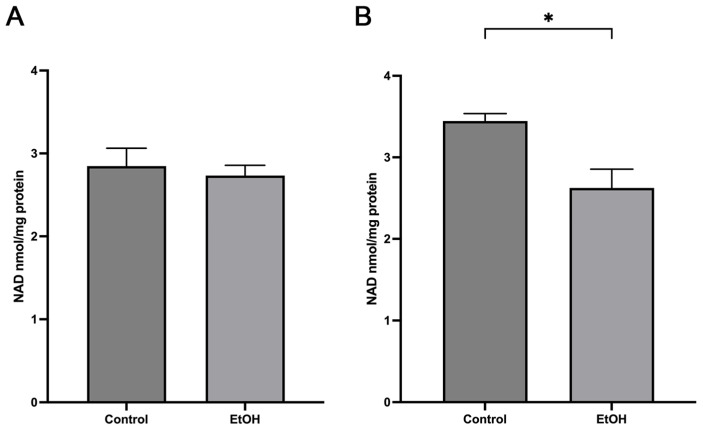
Effect of ethanol metabolism on cerebellar mitochondria NAD levels. (**A**) Ethanol administration did not alter the mitochondria NAD levels isolated from male cerebellum. (**B**) The NAD pools in female mitochondria were significantly reduced one hour following ethanol administration. * *p* < 0.05, n = 4–6.

## Data Availability

Data are available on request.

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
