# Peer review of "Sexual Dimorphism of Ethanol-Induced Mitochondrial Dynamics in Purkinje Cells"

_ijms, 2024, doi:10.3390/ijms252413714_

Round 1

Reviewer 1 Report

Comments and Suggestions for Authors

The manuscript by Khatoon et al. describes a series of elegant experiments aimed at clarification of sexual dimorphism of ethanol-induced alterations in mitochondrial morphology and ROS production in mouse Purkinje cells. To assess alterations in mitochondrial morphology, the authors employed mice expressing mitochondria-targeted yellow fluorescent protein in Purkinje cells in conjunction with confocal microscopy. The authors convincingly demonstrate the stark difference in the effect of ethanol administration on mitochondria in Purkinje cells of male and female mice. While in male mice ethanol does not induce statistically significant alterations, in female mice ethanol administration led to mitochondrial fragmentation, most likely, mediated by activated (phosphorylated) Drp-1. In addition, ethanol induced a decrease in NAD level and increased ROS production in females, but not in males. Consequently, the authors concluded that ethanol induced mitochondrial fragmentation due to elevated production of ROS and Drp-1 phosphorylation in Purkinje cells of female cerebellum. The experiments, described in this manuscript, are well designed and nicely executed. The interpretation of results and conclusions are substantiated by the experimental data. The methods used in this study are sound. Overall, this is a very interesting study shedding light on the important topic of ethanol toxicity and I am confident that this paper will be of great interest to wide audience of neurobiologists and toxicologists.  

Author Response

 We would like to thank the reviewer for appreciation of our work.

Reviewer 2 Report

Comments and Suggestions for Authors

The manuscript of “Sex dimorphism of ethanol-induced mitochondrial dynamics in Purkinje cells” by Rehana Khatoon and co-authors aims to study the changes in mitochondrial dynamics in cerebellar Purkinje cells (PC) after acute ethanol administration. The authors used the unique transgenic mouse model that expresses mitochondria targeted eYFP, driven by Camk2a promotor to visualize mitochondria in Purkinje neurons. The authors found the mitochondria in PC formed an interconnected network of organelles with lengths up to 40 µm. The long tubular mitochondria were identified particularly in the dendritic tree, suggesting Purkinje cell mitochondria could display uniquely long and complex shapes. Moreover, the authors showed for the first time that ethanol metabolism in the cerebellum has different effects on mitochondrial dynamics, with a stronger effect observed in females, which is likely due to a decrease in mitochondrial NAD pools. The latter led to increased superoxide levels and, as a result, to mitochondrial fragmentation.

The manuscript is very interesting and well written. The aim of the study is relevant and timely due to the sharp increase in the number of alcohol-related brain lesions worldwide and the urgent need to find new molecular targets for their effective treatment. The authors focused on an underexplored area of sexually dimorphic effects of alcohol exposure on mitochondrial dynamics in cerebellar Purkinje cells. The research topic is relevant.

The title, abstract, and keywords correspond to the content of the manuscript. The Introduction fully reflects the current state of the issue under study. The methodological approaches are adequate. The purpose of the work is clearly stated in the Introduction. The description of the data analysis is sufficiently detailed. The results are presented in a fairly simple and understandable language, while maintaining clarity, logic and consistency in the presentation of information. The authors' assumptions and conclusions are sufficiently substantiated. Figures are presented in sufficient quantity and clearly reflect the results of the study. The quality of all the figures meets the requirements.

Overall, the research was conducted at a high level, and the manuscript can be accepted in its current form.

Author Response

 We thank the reviewer for his positive comments and for pointing out the novelty of the presented data and importance of studding the underexplored area of sex-dependent effect of ethanol on brain mitochondria metabolism.

Reviewer 3 Report

Comments and Suggestions for Authors

In this manuscript the authors use a transgenic mouse model to visualize and classify the shape of mitochondria in Purkinje cells. They show morphological and biochemical changes in mitochondria from female mice exposed to ethanol, but not from male mice with a similar exposure. They conclude that there is a sex dimorphism of ethanol-induced mitochondrial dynamics in Purkinje cells.

The manuscript is well written and the results are clearly presented. Unfortunately, there are several problems with the data quality and the interpretation of the data:

1) Ethanol exposure
The amount of ethanol intoxication imposed on the mice was very high and very different between males and females. In females, blood levels were 150% of those of males. Overall, the level in females would correspond to 0.8%, a level well above that in which ataxia would emerge (it is 10 times higher compared to the level of 0.08% at which driving is banned in most countries) and coming close to lethal levels. It appears doubtful whether specific conclusions can be drawn from such high exposures.

2) Male / female differences
The main finding of the manuscript is a different outcome between male and female mice. However, it remains unclear whether this difference is simply due to the higher blood levels (145mM vs 103 mM) found in female mice. In order to compare mitochondrial reactions, the experiments would need to be done with similar blood ethanol concentrations in both male and female mice. From the present data it remains unclear whether the observed differences might only be the result of the higher ethanol concentration in female mice.

3) Mitochondrial dynamics
The presented data are not very convincing. On the provided micrographs, single mitochondria cannot be resolved. The authors present a diagram derived from a software which classifies the mitochondria according to size. It is unclear to this reviewer whether these data are reliable, as single mitochondria cannot be resolved in the provided images. Have the results obtained with the software been validated by using an independent procedure? The calculated size of fluorescently labeled objects is strongly dependent on the brightness of the fluorescent label. It seems doubtful that the software can provide reliable data on mitochondrial size, as different labeling intensities might perturb the measurements.

4) Biochemical measurements (Drp1,ROS, NAD)
As in all these measurements, the differences between male and female mice are rather subtle, the question remains whether the observed differences are simply due to the higher ethanol concentration in female mice.

In order to obtain meaningful results, either the experiments with female mice or with male mice need to be repeated with a blood ethanol concentration adjusted to that of the other sex. Even then the meaning of the findings is limited because of the very high ethanol concentration used. In addition, the method for classifying mitochondrial dynamics need to be validated by a an independent method, e.g. electron microscopy.

Author Response

 In this manuscript the authors use a transgenic mouse model to visualize and classify the shape of mitochondria in Purkinje cells. They show morphological and biochemical changes in mitochondria from female mice exposed to ethanol, but not from male mice with a similar exposure. They conclude that there is a sex dimorphism of ethanol-induced mitochondrial dynamics in Purkinje cells.

The manuscript is well written and the results are clearly presented. Unfortunately, there are several problems with the data quality and the interpretation of the data:

  • Ethanol exposure 
    The amount of ethanol intoxication imposed on the mice was very high and very different between males and females. In females, blood levels were 150% of those of males. Overall, the level in females would correspond to 0.8%, a level well above that in which ataxia would emerge (it is 10 times higher compared to the level of 0.08% at which driving is banned in most countries) and coming close to lethal levels. It appears doubtful whether specific conclusions can be drawn from such high exposures.

For clarity, the amount of ethanol administered was the same for males and females. The reviewer’s point is taken, however, that the different level of BAL detected at our timepoint(s) may account for the observed sex differences rather than a sex difference in response to ethanol per se.  To our knowledge, many published papers in which acute ethanol effect was studied in mouse used 2 to 6 g/kg of ethanol by i.p. injections (Jin S. et al. ,Nature, 2021; Mansouri A., J. Pharmacology and Exp. Therapeutics, 2001; Escarabajal et al., Pharmacology 2002; Hasegawa et al., Biol. Pharm. Bull. 2023; Wilson et al., Frontier in Neuroscience; 2023, Jung and Metzger, Molecules 2010; Karadayian et al., Neuroscience 2015; Light et al., Neuroscience 2002). Thus, we used the lowest dose that was used in the published literature (2 mg/kg). The reason for using this “high” dose of ethanol (when compared to humans) is that the mouse metabolism is up to 10-fold faster than humans. Our data detect the different rate of ethanol metabolism between male and female animals.  This is also observed in humans (Dettling A. et al. Alcohol 2007).

  • Male / female differences
    The main finding of the manuscript is a different outcome between male and female mice. However, it remains unclear whether this difference is simply due to the higher blood levels (145mM vs 103 mM) found in female mice. In order to compare mitochondrial reactions, the experiments would need to be done with similar blood ethanol concentrations in both male and female mice. From the present data it remains unclear whether the observed differences might only be the result of the higher ethanol concentration in female mice.

We do agree that one of the possible explanations for the differences in the ethanol effect between the male and females is the higher blood levels in females compared to males. However, this is due to difference in the rate of ethanol metabolism between males and females. We have modified the discussion accordingly.

  • Mitochondrial dynamics
    The presented data are not very convincing. On the provided micrographs, single mitochondria cannot be resolved. The authors present a diagram derived from a software which classifies the mitochondria according to size. It is unclear to this reviewer whether these data are reliable, as single mitochondria cannot be resolved in the provided images. Have the results obtained with the software been validated by using an independent procedure? The calculated size of fluorescently labeled objects is strongly dependent on the brightness of the fluorescent label. It seems doubtful that the software can provide reliable data on mitochondrial size, as different labeling intensities might perturb the measurements.

We do not agree with the reviewer. In fig 2 one can readily resolve many individual mitochondria. Additionally, we marked the single axonal mitochondria with triangle.  Only the perinuclear mitochondria are not possible to resolve, however, these were not included into quantification as stated in the Methods section.

            We have published several papers using these transgenic animals to quantify mitochondria fragmentation in brain following ischemic insult (Owens K. et al., J Bioenerg Biomembr 2014; Long A., BMC Neurology 2015; Klimova N., J Neuroscience Research 2019; Klimova N. Exp Neurology 2020). Similar results of fragmented mitochondria in ischemic brain tissue were reported by other labs using TEM (Kumar R., Mol Cell Neurosci 2016; Solenski N. Stroke, 2002, Uchino H., Neurobiology Dis 2002).

4) Biochemical measurements (Drp1,ROS, NAD)
As in all these measurements, the differences between male and female mice are rather subtle, the question remains whether the observed differences are simply due to the higher ethanol concentration in female mice.

In order to obtain meaningful results, either the experiments with female mice or with male mice need to be repeated with a blood ethanol concentration adjusted to that of the other sex. Even then the meaning of the findings is limited because of the very high ethanol concentration used. In addition, the method for classifying mitochondrial dynamics need to be validated by an independent method, e.g. electron microscopy.

The point of presented experiments is to compare the ethanol effects on brain mitochondria in male and female animals by administering the same ethanol amount per weight to both males and females.

Round 2

Reviewer 3 Report

Comments and Suggestions for Authors

The authors have responded to the criticism of the reviewer, but have not substantially changed the manuscript. In its present form, the major problems with the manuscript are unchanged and the remarks form the original review are still valid. Of course, the authors have a different view about these points but serious doubts about the meaning of the presented data remain.

Author Response

The authors have responded to the criticism of the reviewer, but have not substantially changed the manuscript. In its present form, the major problems with the manuscript are unchanged and the remarks form the original review are still valid. Of course, the authors have a different view about these points but serious doubts about the meaning of the presented data remain.

We substantiated our previous responses by supporting them with several references to published data. Thus, these are not our views or opinions.